# SCL14 Inhibits the Functions of the NAC043–MYB61 Signaling Cascade to Reduce the Lignin Content in Autotetraploid *Populus hopeiensis*

**DOI:** 10.3390/ijms24065809

**Published:** 2023-03-18

**Authors:** Jian Wu, Bo Kong, Qing Zhou, Qian Sun, Yaru Sang, Yifan Zhao, Tongqi Yuan, Pingdong Zhang

**Affiliations:** 1National Engineering Research Center of Tree Breeding and Ecological Restoration, Beijing Forestry University, Beijing 100083, China; 2Key Laboratory of Genetics and Breeding in Forest Trees and Ornamental Plants, Ministry of Education, Beijing Forestry University, Beijing 100083, China; 3College of Biological Sciences and Technology, Beijing Forestry University, Beijing 100083, China; 4Beijing Key Laboratory of Lignocellulosic Chemistry, Beijing Forestry University, Beijing 100083, China

**Keywords:** autotetraploid, lignin biosynthesis, plant hormone, differentially expressed gene, transcription factor

## Abstract

Whole-genome duplication often results in a reduction in the lignin content in autopolyploid plants compared with their diploid counterparts. However, the regulatory mechanism underlying variation in the lignin content in autopolyploid plants remains unclear. Here, we characterize the molecular regulatory mechanism underlying variation in the lignin content after the doubling of homologous chromosomes in *Populus hopeiensis*. The results showed that the lignin content of autotetraploid stems was significantly lower than that of its isogenic diploid progenitor throughout development. Thirty-six differentially expressed genes involved in lignin biosynthesis were identified and characterized by RNA sequencing analysis. The expression of lignin monomer synthase genes, such as *PAL*, *COMT*, *HCT*, and *POD*, was significantly down-regulated in tetraploids compared with diploids. Moreover, 32 transcription factors, including MYB61, NAC043, and SCL14, were found to be involved in the regulatory network of lignin biosynthesis through weighted gene co-expression network analysis. We inferred that SCL14, a key repressor encoding the DELLA protein GAI in the gibberellin (GA) signaling pathway, might inhibit the NAC043–MYB61 signaling functions cascade in lignin biosynthesis, which results in a reduction in the lignin content. Our findings reveal a conserved mechanism in which GA regulates lignin synthesis after whole-genome duplication; these results have implications for manipulating lignin production.

## 1. Introduction

Lignin is a principal structural component in the secondary cell walls (SCWs) of plant cells and the second most abundant plant lignocellulosic material in nature [1,2]. It enhances the mechanical strength of the cell wall, facilitates water transport, and often functions as a physical barrier against pathogen attack [3]. However, it is interwoven with cellulose and hemicellulose in the SCW, which forms a structural matrix around the cellulose microfibrils; it is considered an important factor limiting the depolymerization and utilization of plant lignocellulosic resources [4,5,6]. Hence, plants with a low lignin content are superior for lignocellulosic bioenergy production, pulp and paper making, and forage digestion.

Polyploidy, or whole-genome duplication, has long been recognized as a major force in angiosperm evolution [7,8]. According to the origin of the genome duplication event, polyploids can be classified as allopolyploids (combinations of two or more divergent genomes) and autopolyploids (multiplications of a single genome) [9,10]. The combination of distinct genomes in allopolyploids can result in stochastic changes in genome organization, the differential contribution of parental genomes, and additive heterosis effects [9]. Allopolyploids can be confounded by the entanglement of both genome duplication and hybridization. In contrast, autopolyploids exhibit only gene dosage effects associated with the doubling of their genetic information [11]. Therefore, synthesized autopolyploids provide ideal systems for investigating the mechanisms underlying gene dosage effects on trait variation due to their relatively uniform genetic background.

Whole-genome duplication events often confer autopolyploid plants with superior traits compared with those of their diploid counterparts. Several studies have shown that increases in somatic ploidy can alter the composition of the cell wall in a way that might be beneficial for the production of bioenergy and materials. For example, Corneillie et al. [12] showed that the somatic ploidy level is negatively correlated with the lignin content in *Arabidopsis* stem tissue; the lignin content is 20%, 50%, and 55% lower in tetraploids, hexaploids, and octoploid than in diploids, respectively. Serapiglia et al. [13] found that the lignin content in shrub willow tends to be lower in tetraploids than in diploids. A similar phenomenon has been observed in tetraploid crops and horticultural plants such as rice straw [14], potato straw [15], and *Trifolium pratense* [16]; the lignin content of tetraploids is lower than that of diploids. These studies have provided useful insights into the effects of whole-genome duplication on lignin content variation. However, the effects of whole-genome duplication on lignin biosynthesis, and its underlying regulatory mechanisms, remain unclear.

Here, we characterized the molecular regulatory mechanism underlying lignin content variation after the doubling of homologous chromosomes in *Populus hopeiensis*. For this, we used diploid and autotetraploid *P. hopeiensis* as study materials to analyze their growth trait, cell wall thickness, lignin content, hormone content, and transcription sequencing. Further, the expression patterns of key genes involved in lignin biosynthesis pathways between diploids and tetraploids were specifically investigated. The hub transcription factors (TFs) involved in the regulatory network of lignin biosynthesis were identified through weighted gene co-expression network analysis (WGCNA). Our findings enhance our understanding of how autotetraploidization improves the utility of plants for the bio-based economy by altering the lignin content.

## 2. Results

### 2.1. Variation in Growth, Cell Wall Characteristics, Lignin Content, and Hormone Content in Diploid and Tetraploid Plants

As shown in Figure 1, diploid and tetraploid *P. hopeiensis* grew rapidly at 3 and 6 months of age, and sharp basal diameter and height increases were observed. At 9 months of age, the leaves at the bottom of the diploid and tetraploid plants began to turn yellow. At 12 months of age, the leaves of the diploid and tetraploid plants completely dropped and stopped growing.

The basal diameter and height of diploid and tetraploid plants are shown in Appendix A. The mean basal diameter and height of the tetraploids of the three clones were significantly less than that of diploids at all growth stages (*t*-test). The growth rate of tetraploid plants was lower than that of diploid plants, and differences in basal diameter and height increased with plant development. Pronounced differences in growth were observed among the three clones; the growth of the ramets of clone BT1 was the fastest.

Changes in the cell wall characteristics in diploid and tetraploid *P. hopeiensis* stems of the three clones at four different developmental stages were characterized (Figure 2). The cell walls were thicker at all growth stages, and the cell lumen was larger in tetraploids than in their diploid counterparts. The cell wall gradually enlarged and thickened from 3 months of age to 12 months of age, indicating that the degree of lignification increased with stem tissue development. The cell wall thickness in tetraploids of the three clones was significantly higher than that of diploids at all developmental stages (Figure 3A, *t*-test). The lignin content of diploid and tetraploid plants of the three clones at the four developmental stages was further analyzed (Figure 3B). The lignin content was significantly lower in tetraploids than in diploids (*t*-test). The lignin content of clones BT1, BT3, and BT8 tetraploids was decreased by 3.9–8.4%, 4.7–9.3%, and 6.2–10.5% at four developmental stages, respectively. These results indicate that whole-genome duplication reduced the lignin content in autotetraploid plants compared with their diploid counterparts [12,13,14,15,16]. The lignin content of diploid and tetraploid plants increased with growth. Changes in the lignin content and cell wall thickness in diploids and tetraploids of the three clones were similar during the growing season. Hence, only diploid and tetraploid plants of the BT1 clone were randomly sampled for transcriptome sequencing.

To characterize differences in the content of plant hormones in plants with different ploidy levels, the content of plant hormones, such as gibberellin (GA), brassinosteroids (BR), jasmonic acid (JA), auxin (IAA), and abscisic acid (ABA), was measured in diploid and tetraploid plants of the three clones at the four growth stages (Appendix A). The GA content was significantly lower in tetraploids than in diploids. The GA content of clones BT1, BT3, and BT8 tetraploids decreased by 13.3–22.9%, 12.8–20.2%, and 9.9–15.0% at the four growth stages, respectively. The IAA content of the 3-, 6- and 9-month-old tetraploids of clones BT1 and BT8 was significantly lower than that of the diploids. The IAA content of clones BT1, BT3, and BT8 tetraploids were decreased by 7.5–19.3%, 3.3–14.6%, and 7.8–15.0% at four developmental stages, respectively. However, the BR content of tetraploids was significantly higher than that of diploids. The BR content of clones BT1, BT3, and BT8 tetraploids were increased by 8.6–17.3%, 7.2–13.1%, and 11.5–17.6% at four developmental stages, respectively. The JA content of tetraploids was significantly higher than that of diploids at the four growth stages, with the exception of clone BT8 at 12 months of age. The JA content of clones BT1, BT3, and BT8 tetraploids were increased by 12.3–20.9%, 10.9–15.9%, and 8.9–12.9% at the four growth stages, respectively. There was no significant difference in the ABA content between diploid and tetraploid plants in early development, and only the ABA content was significantly higher than that in diploids at 12 months of age. The content of BR and JA increased, and the content of GA decreased with development. The content of IAA first increased and then decreased, and the content of ABA first decreased and then increased. Phenotypic correlations between each clone’s GA and lignin content at all developmental stages were investigated (Appendix A). A significant positive correlation between the GA and lignin content of the three clones was observed at each developmental stage, indicating that GA might play an important role in regulating lignin biosynthesis in *Populus*.

### 2.2. Analysis of Transcriptional Profiles between Diploids and Tetraploids

To clarify the genetic control of lignin biosynthesis in tetraploid *Populus*, 24 stem tissue samples were randomly collected for transcriptome sequencing at 3, 6, 9, and 12 months of age. A total of 160.5 Gb of clean data were obtained. The Q20, Q30, and GC content of the clean reads were greater than 97.6%, 93.5%, and 44.4%, respectively (Appendix A), suggesting that the RNA-seq data were of high quality. The transcriptome data were mapped to the *Populus tomentosa* genome, and the mapping efficiency ranged from 82.8–93.0% (Appendix A).

To clarify the effects of genome doubling on the expression patterns of DEGs, a comparative transcriptomic analysis was performed using DEseq2 software (version 1.20.0) (Appendix A). A total of 2243 significant DEGs were identified, including 1232 DEGs in D3 vs. T3, 264 DEGs in D6 vs. T6, 523 DEGs in D9 vs. T9, and 391 DEGs in D12 vs. T12 (Figure 4A). After genome doubling, gene expression levels changed in every sample at all growth stages, especially at 3 months of age. DEGs observed after genome doubling were massively up-regulated in the early growth stages of *P. hopeiensis* (Figure 4B). To determine the reliability of the RNA-seq data, a total of eight candidate genes were selected and validated by quantitative real-time PCR (qRT-PCR) analysis (Appendix A). The expression patterns of these genes in the qRT-PCR analysis were consistent with the transcriptome sequencing data.

### 2.3. Functional Analysis of DEGs between Diploids and Tetraploids

To further clarify the functions of DEGs that were observed after genome doubling, GO analysis of DEGs in D3 vs. T3, D6 vs. T6, D9 vs. T9, and D12 vs. T12 was conducted. At all growth stages, the first three terms in the biological process category were metabolic, cellular, and single-organism. The main terms in the cellular component category were membrane, membrane part, cell, and cell part. Most genes in the molecular function category were enriched in binding and catalytic activity (Figure 5A).

GO and KEGG enrichment analyses of all DEGs between diploids and tetraploids were performed. The most significantly enriched GO terms included TF activity and peroxidase activity (Figure 5B, Appendix A). In the KEGG enrichment analysis, DEGs between diploids and tetraploids were significantly (*q*-value < 0.05) enriched in the pathways of phenylpropanoid biosynthesis and plant hormone signal transduction (Figure 5C, Appendix A). DEGs between diploids and tetraploids were involved in lignin biosynthesis and plant hormone signaling pathway.

### 2.4. Identification and Expression Analysis of DEGs Involved in Lignin Biosynthesis

Lignin synthesis is one of the most important pathways in phenylpropanoid biosynthesis, and many enzymes and genes participate in this process. We identified the DEGs that encode enzymes involved in the lignin biosynthesis pathway of *P. hopeiensis*. A total of 36 DEGs across eight gene families were identified as enzyme-coding genes involved in lignin biosynthesis (Appendix A).

The expression levels of DEGs involved in the lignin biosynthesis pathway were further analyzed (Figure 6). Most genes were highly expressed at 3, 6, and 9 months of age and weakly expressed at 12 months of age. Phenylalanine ammonia-lyase (*PAL*) is the first rate-limiting enzyme in the entire pathway. The expression of two *PAL* genes was significantly down-regulated in 9-month-old tetraploids relative to diploids. Hydroxycinnamoyl-CoA shikimate (*HCT*) is also a key rate-limiting enzyme in lignin biosynthesis. The expression of two *HCT* genes was down-regulated in 3-month-old and 9-month-old tetraploids. The expression of one caffeic acid *O*-methyltransferase (*COMT*) gene was down-regulated in 6-month-old tetraploids, and the expression of one *COMT* gene was down-regulated in 12-month-old tetraploids. The expression of one cinnamyl alcohol dehydrogenase (*CAD*) gene was down-regulated in 3-month-old tetraploids. A total of 19 peroxidases (*POD*) genes were differentially expressed between diploids and tetraploids at all growth stages. The expression of six *POD* genes was down-regulated in 3-month-old tetraploids, the expression of four *POD* genes was down-regulated in 9-month-old tetraploids, and the expression of eight *POD* genes was up-regulated in 6-month-old tetraploids. These results indicate that the expression of most lignin biosynthesis genes was down-regulated in 3- and 9-month-old tetraploids; this might be related to decreases in the lignin content [2,3].

### 2.5. Identification of Plant Hormone Signal Transduction Genes and TFs

To further clarify the functions of phytohormones involved in lignin biosynthesis, a total of 94 DEGs involved in plant hormone signaling pathways were obtained (Appendix A), and the FPKM value of each gene was used to build a heatmap (Figure 7). In the GA signaling pathway, the expression of 19 DELLA repressors was up-regulated in 3-month-old tetraploids, and the expression of three DELLA repressors was up-regulated in 9-month-old tetraploids. However, the expression of one GID1 receptor was significantly down-regulated in 3-month-old tetraploids. In the BR signaling pathway, 19 BAK1 and 16 BRI1 receptor expression were substantially up-regulated in tetraploids at all growth stages. The expression of two BZR1/2 TFs was significantly up-regulated in 3-month-old tetraploids. In the JA signaling pathway, the expression of five JAZ repressor proteins was up-regulated in 3-, 6-, or 12-month-old tetraploids, and the expression of nine MYC2 TFs was significantly up-regulated in 3-month-old tetraploids. In the auxin signaling pathway, the expression of three AUX/IAA transcription repressors was up-regulated in 3-, 6-, and 9-month-old tetraploids. In the ABA signaling pathway, the expression of two SNRK2 kinases was significantly up-regulated in 12-month-old tetraploids. These pathways and their corresponding DEGs might be key factors affecting differences in the lignin content between tetraploids and diploids.

TFs play important roles in regulating lignin biosynthesis. To explore the roles of TFs in lignin biosynthesis, DEGs encoding TFs in diploids and tetraploids were predicted (Appendix A). A total of 261 DEGs in 34 TF families were identified, and the most common TF families observed were WRKY, AP2/ERF-ERF, GRAS, NAC, MYB, bHLH, and C2H2 (Appendix A). A total of 183 DEGs in 29 TF families, 22 DEGs in 11 TF families, 30 DEGs in 16 TF families, and 54 DEGs in 18 TF families were identified in 3-, 6-, 9-, and 12-month-old plants, respectively.

### 2.6. Co-Expression Network Analysis

To identify the regulatory relationships between TFs and lignin biosynthesis-related genes, a co-expression network was constructed using WGCNA. A total of 12 modules were identified, and genes with kME > 0.7 were selected as the members in each module (Appendix A). Genes in the brown module were significantly enriched in cell wall organization and plant-type SCW biogenesis in the GO database; these same genes were enriched in phenylpropanoid biosynthesis and plant hormone signal transduction in the KEGG database (Appendix A). Genes in the orange module were significantly enriched in TF activity and DNA binding in the GO database; these same genes were enriched in phenylpropanoid biosynthesis and plant hormone signal transduction in the KEGG database (Appendix A). These results indicate that genes in these two modules, especially the brown module, might play important roles in lignin biosynthesis.

The gene co-expression network of the top 16 TFs highly correlated with six lignin structural genes in the brown module was constructed (Figure 8A and Appendix A). The hub genes with the highest connectivity in the center of the regulatory network were MYB61 (gene-POTOM_018306) and NAC043 (gene-POTOM_008542). These two hub genes were highly co-expressed with *POD* (*Populus*_*tomentosa*_new Gene_525), which is involved in lignin biosynthesis [2,3]. The gene co-expression network of the top 16 TFs significantly correlated with five lignin structural genes in the orange module was also constructed (Figure 8B and Appendix A). The hub genes in the orange module were identified as SCL14 (gene-POTOM_001873) and CIGR1 (gene-POTOM_039977). SCL14 and CIGR1 encoded the DELLA protein GAI in the GA signaling pathway and showed high connectivity with *PER72* (Populus_tomentosa_newGene_1720).

qRT-PCR was performed to detect the expression patterns of four TFs and two lignin structural genes (Figure 9). The expressions of MYB61, NAC043, and *POD* first increased and then decreased with development; the expression of these genes was highest at 6 months of age. The expression of these three genes was significantly down-regulated in 3- and 6-month-old tetraploids, which likely means that MYB61 and NAC043 act as positive regulators of lignin formation during early growth. The expression of SCL14 and CIGR1 was significantly up-regulated, and the expression of *PER72* was significantly down-regulated in 3- and 6-month-old tetraploids. The expression of these three genes was significantly lower in 6-month-old plants than in 3- and 9-month-old plants, which suggests that SCL14 and CIGR1 encoded repressors involved in lignin biosynthesis during early growth. The opposite expression patterns were observed for NAC043 and SCL14 in 3-, 6-, and 9-month-old plants, which suggests that these two sets of genes mutually inhibit one another’s expression.

## 3. Discussion

### 3.1. Whole-Genome Duplication Mediated Reductions in the Lignin Content

Gene duplication events, especially whole-genome duplications, provide genomic plasticity for the functional divergence of replicated genes, genomic and/or chromosomal recombination, transcriptome changes, and gene dosage effects, contributing to phenotypic variation [7,8]. The lignin content is usually low in autopolyploid plants, such as *Arabidopsis* [12], *Miscanthus sacchariflorus* [17], and *Trifolium pratense* [16]. Our findings indicate that the lignin content of tetraploids *P. hopeiensis* was significantly lower than that of their diploid counterparts. This suggests that gene dosage effects might increase the size of cells and organelles by altering gene expression, and this is usually accompanied by a decrease in cell surface area per unit volume, which might reduce lignin accumulation [18]. We also found that the cell wall thickness of tetraploids was significantly higher than that of diploids, indicating that the reduction in the lignin content in the cell wall might be compensated for by another cell wall polymer or matrix polysaccharide, such as cellulose, hemicellulose, and pectin [12]. These results indicate that tetraploids might employ a specific mechanism for regulating lignin biosynthesis based on their own needs, and this might be achieved by reducing the expression of lignin biosynthesis genes and altering the expression of TFs.

The lignin biosynthesis pathway is a complex process involving many biological events that are strictly regulated by various genes [19]. Lignin monomer synthase genes, including *PAL*, *4CL*, *COMT*, *HCT*, *CSE*, *CCR*, *CAD*, and *POD*, have been investigated in *Arabidopsis thaliana* [20,21], *Populus* [22], and *Cunninghamia lanceolata* [23]. *PAL* is the first enzyme-encoding gene involved at the beginning of this pathway, and it determines the metabolic flow of the entire pathway [24]. *HCT* is a key rate-limiting enzyme-encoding gene involved in the synthesis of G and S monomers [25]. *COMT* regulates the formation of ferulic acid, sinapic acid, coniferyl-aldehyde, and sinapaldehyde [25]. In addition, the final stage of lignin polymerization requires the oxidation of monolignols by *POD*, which initiates random cross-linking to form the lignin polymer [26]. In this study, the expression of most of the DEGs involved in the lignin biosynthesis pathway was significantly down-regulated in tetraploids at 3 or 9 months of age relative to diploids, especially at 3 months of age (Figure 6). Thus, the down-regulated expression of key genes in the lignin biosynthesis pathway during early and late growth might lead to decreases in the lignin content of tetraploids [20,21,22].

### 3.2. Whole-Genome Duplication Mediated the Regulation of Plant Hormones

Aside from a shift in the composition of the cell wall, whole-genome duplication can affect the content of plant hormones [27,28]. We found that the content of hormones such as GA, IAA, BR, and JA in the induced tetraploids significantly differed from that in diploids, and the lignin content was significantly positively correlated with the GA content, suggesting that GA might play an important role in regulating lignin biosynthesis after whole-genome doubling.

GAs are plant-growth-promoting hormones that play key roles in diverse aspects of plant growth and development [29]. The primary roles of GAs during wood formation include regulating the early stages of wood differentiation, such as cell elongation and expansion [30] and differentiation of the xylem from the cambium [31]. The binding of GA to the GID1 receptor stimulates the formation of the GID1-GA-DELLA complex, which triggers DELLA protein ubiquitination and proteolysis [32]. DELLA proteins in the GRAS gene family are the major repressors of GA signaling [33]. DELLAs have been shown to inhibit GA-promoted growth by interacting with key regulatory proteins and blocking their DNA-binding or transactivation activities [34]. Interactions between DELLA proteins and NAC genes have recently been shown to mediate GA signaling to inhibit lignin biosynthesis in cotton stems [29]. In this study, the expression of one GID1 gene (*CXE15*) was down-regulated in tetraploids, and 20 DELLA genes were highly expressed in tetraploids (Figure 7), which suggests that reductions in the concentration of GA abolished DELLA repression, promoted the activities of downstream proteins repressed by DELLAs, and inhibited lignin biosynthesis [29,30,31]. Hence, we speculate that DELLA proteins are key specific regulators controlling lignin biosynthesis. In this study, SCL14, an ortholog of *Arabidopsis* GAI, strongly interacted with enzyme-coding genes involved in lignin biosynthesis (Figure 8B), which is consistent with the results of our study. Furthermore, our results suggest that tetraploids with a reduced content of GA exhibit a dwarf phenotype, which is consistent with the findings of previous studies [28].

In addition to GAs, other plant hormones are involved in lignin formation. AUX/IAA transcription repressors inhibit the activity of ARF proteins in the auxin signaling pathway [35,36]. In poplar, overexpression of *PtrARF2.1* results in the activation of many genes involved in lignin biosynthesis [36]. In our study, the expression of three IAA14 proteins was significantly up-regulated in tetraploids; these proteins inhibited the activity of two *ARF* genes (*ARF11* and *ARF15*), which likely contributed to decreases in the lignin content.

In the BR signaling pathway, BZR1/2 is de-phosphorylated and activated to regulate the expression of a suite of downstream TFs and genes [37,38,39]. A recent study has shown that PbBZR1 represses the expression of lignin biosynthesis genes in *Pyrus bretschneideri* [40]. The expression of two BZR1 proteins (BZR1 and BEH4) was significantly up-regulated in tetraploids, suggesting that BZR1 might negatively regulate lignin biosynthesis.

The JAZ protein, a key repressor of JA signaling, suppresses the transcription of JA-responsive genes by interacting with MYC2 [41]. Previous studies have shown that DELLAs prevent inhibitory interactions of JAZ with MYC2 and promote JA signaling; this might be the mechanism by which GA mediates the down-regulation of JA defense responses [42,43,44]. However, this defense response is costly and often accompanied by significant growth inhibition [45]. DELLAs interact with ARF and BZR1 to mediate crosstalk between GA, IAA, and BR [46,47,48,49,50]. Therefore, DELLAs might be key components for integrating GA with other phytohormones in the regulation of wood formation.

### 3.3. Transcription Regulatory Modules Involved in Lignin Biosynthesis

TFs play an important role in plant growth and development [51,52,53]. NAC TFs, considered one of the main regulators of wood formation, are switches involved in SCW formation [51,52]. In *Arabidopsis*, a multilevel regulatory network consisting of NAC and MYB TFs has been shown to regulate lignin biosynthesis [52,53]. The top-level secondary wall NAC (VND1-7, NST1-2, and SND1) master switches can directly activate the expression of downstream TFs, such as MYB family members, and together they can regulate the expression of lignin biosynthesis genes [51,53]. In this study, NAC043, a homolog of *Arabidopsis* NST1, was found to be highly co-expressed with enzyme-coding genes involved in lignin biosynthesis (Figure 8A). NST1 is a key regulator involved in forming secondary walls in the woody tissues of A. thaliana that activates a cascade of downstream TFs that activate pathways that mediate the biosynthesis of individual SCW components, such as cellulose, xylan, and lignin [54,55]. Previous studies have shown that the inhibition of NST1 expression in *Medicago truncatula* results in the reduced expression of lignin biosynthesis pathway genes [56]. Furthermore, CgNAC043, the *Citrus grandis* ortholog of *Arabidopsis* NST1, is the master switch that directly activates the expression of many lignin biosynthesis genes [57]. Therefore, NAC043 might be the master switch that activates the expression of downstream TFs and lignin biosynthesis genes in this network.

MYB TFs are one of the largest TF families in higher plants and play key roles in regulating the phenylpropanoid pathway and the biosynthesis of lignin [21,58]. In this study, MYB61*,* which is predicted to be an ortholog of *Arabidopsis* AtMYB61, was found to be highly co-expressed with enzyme-coding genes involved in lignin biosynthesis (Figure 8A). AtMYB61, which encodes a member of the *A. thaliana* R2R3-MYB family of TFs, is a transcription activator that regulates lignin biosynthesis [58,59]. Previous studies have shown that PtMYB8, the *Pinus taeda* ortholog of *Arabidopsis* AtMYB61, is a positive regulator of lignin biosynthesis [60]. In addition, PtoMYB74, a homolog of *Arabidopsis* AtMYB61, has been shown to activate the expression of structural genes and positively regulate lignin biosynthesis in *A. thaliana* and *P. tomentosa* [59]. Therefore, MYB61 might regulate lignin biosynthesis by modulating the expression of downstream TFs in the multilevel regulatory network.

A recent study has shown that a DELLA protein (GAI) interacts directly with a wide range of SCW-related NAC TFs, which are homologous to most switches that promote SCW formation, and induces the silencing of these NAC genes to inhibit SCW development by down-regulating lignin biosynthesis and deposition in cotton stems [29]. Previous studies have shown that a conserved GA-mediated DELLA–NAC signaling cascade regulates SCW biosynthesis, and GA signals promote SCW synthesis by relieving the interaction between SLENDER RICE1 (SLR1), a key repressor encoding the DELLA protein, and NAC29/31, the top-layer master switches for the transcriptional regulation of secondary wall formation, which promotes the expression of the activator MYB61 in rice [34]. This signaling cascade is regulated by endogenous GA levels and is required for internode development [34]. These results show that DELLAs might interact with multiple SCW regulators and comprise a complex regulatory network that controls GA signaling during SCW development. In our study, the top-level secondary wall gene NAC043 acts as the master switch that directly activates the expression of MYB61, which in turn regulates the expression of lignin structural genes. However, this hierarchical regulatory pathway is blocked by SCL14–NAC043 interactions. Analysis of expression patterns revealed that SCL14, a key repressor encoding the DELLA protein GAI in the GA signaling pathway, inhibits the activity of NAC043 during lignin formation (Figure 9). Therefore, SCL14 and NAC043 comprise a transcriptional regulatory network mediating the regulation of lignin biosynthesis by GA in *P. hopeiensis* stems (Figure 10). This suggests that whole-genome doubling leads to a decrease in lignin. However, further functional validation is needed.

## 4. Materials and Methods

### 4.1. Plant Materials

The National Engineering Laboratory for Tree Breeding, Beijing Forestry University, China, supplied tissue culture plantlets of the diploid and autotetraploid *P. hopeiensis* clones BT1, BT3, and BT8 [61]. The sterile-rooted plantlets were introduced into a solid rooting medium supplemented with half-strength Murashige and Skoog (MS) medium [62], 3% (*w*/*v*) sucrose, 0.65% (*w*/*v*) agar, and 0.2 mg/L IBA. All media were adjusted to pH 5.8–6.2 before autoclaving for 15 min at 121 °C. All cultures were illuminated by fluorescent tubes with a 14-h photoperiod at 2000 lx in a growth chamber at 25 °C.

### 4.2. Morphological Characteristics and Sample Preparation

After rooting culture for 30 days, 45 diploids and 45 tetraploids of each clone with prolific root systems were selected, transplanted into plastic pots filled with potting media (turfy soil:perlite:vermiculite, 2:1:1), and grown in a greenhouse. After 3, 6, 9, and 12 months of culture, the basal diameter and plant height of the diploid and tetraploid plants of the three clones were recorded. Next, the main stems of diploid and tetraploid plants of the three clones at four growth stages were collected for cell wall analysis and lignin content determination. Segments (2 cm) from the middle of the stem were rapidly harvested, frozen in liquid nitrogen, and then stored in a freezer at −80 °C for hormone content determination, transcriptome sequencing, and qRT-PCR analysis verification. Three biological replicates were performed for samples in each stage.

### 4.3. Measurement of Cell Wall Thickness, Lignin Content, and Hormone Content

Stem samples of diploid and tetraploid plants of the three clones at the four growth stages were used for measurements of cell wall thickness, lignin content, and hormone content. Transverse sections from the base of the stem were cut and frozen in liquid nitrogen; they were then dried in a freeze-dryer (DC401, Yamato, Tokyo, Japan) for 2 days. The samples were sputter-coated with a thin layer of platinum (MCLOOO, Hitachi, Tokyo, Japan) for 30 s and observed at an accelerating voltage of 3 kV using a field emission scanning electron microscope (FE-SEM) (SU8010, Hitachi, Japan). The cell wall thickness was determined using ImageJ software (version 1.8.0) (http://rsb.info.nih.gov/ij/, accessed on 21 January 2022).

The lignin content was determined following the National Renewable Energy Laboratory standard analytical procedure of the United States [63]. The lignin content was equal to the sum of the acid-insoluble lignin content and acid-soluble lignin content. The content of GA, BR, JA, IAA, and ABA was determined using enzyme-linked immunosorbent assays by Shanghai Enzymatic Biotechnology Company Ltd. (Shanghai, China).

### 4.4. RNA-Seq and Mapping

The stem samples of diploid and tetraploid plants of the clone BT1 at ages of 3, 6, 9, and 12 months were used for RNA-seq, with the diploid material denoted as D3, D6, D9, and D12, and the tetraploid material denoted as T3, T6, T9, and T12, respectively. Total RNA was extracted from the stem samples, and the concentration and purity of RNA were measured using the NanoDrop 2000 spectrophotometer (Thermo Fisher Scientific, Wilmington, DE, USA). The RNA integrity was verified using the RNA Nano 6000 Assay Kit of the Agilent Bioanalyzer 2100 system (Agilent Technologies, Santa Clara, CA, USA). Following the manufacturer’s recommendations, the cDNA library was constructed using the NEBNext UltraTM RNA Library Prep Kit for Illumina (NEB, Ipswich, MA, USA). Transcriptome sequencing was performed on an Illumina NovaSeq 6000 sequencing platform to generate paired-end raw reads, and the raw data were used in subsequent analyses.

To obtain high-quality clean data, the raw data were filtered by removing all adapters, poly-N-containing sequences, and low-quality reads. The Q20, Q30, GC content and sequence duplication levels of the clean data were determined to evaluate sequencing quality. The clean reads were mapped to the genome database of *P. tomentosa* using HISAT2 software (version 2.1.0) [64], and the alignment rate was calculated. FPKM values were calculated to evaluate the expression levels of genes in samples [65].

### 4.5. Identification of DEGs and Functional Analysis

Differential expression analysis of diploid and tetraploid plants at four growth stages was performed using the DESeq2 package [66] with the following screening criteria: fold change ≥ 1.5 and a false discovery rate < 0.05. Gene Ontology (GO) (http://geneontology.org/, accessed on 11 March 2022) enrichment analysis of DEGs was performed using the GOseq R package based on Wallenius’ non-central hypergeometric distribution [67]. Furthermore, the Kyoto Encyclopedia of Genes and Genomes (KEGG) (http://www.kegg.jp/, accessed on 11 March 2022) pathway analysis of DEGs was conducted using KOBAS 2.0 software [68]. TFs from DEGs were predicted using the PlantTFDB [69].

### 4.6. Gene Co-Expression Network Analysis and Visualization

WGCNA for all expressed genes (FPKM > 1) of diploid and tetraploid plants at four growth stages was implemented using an R package [70]. The modules were acquired through the automatic network established function (blockwise modules), and genes with kME > 0.7 were designated as members of the module. The modules most strongly related to the lignin content (r < −0.4 or > 0.4 and *p* < 0.05) were selected for GO and KEGG functional enrichment analyses. Gene co-expression networks were visualized using Cytoscape software (version 3.8.2) [71]. The hub genes in the modular network were distinguished and ranked based on their expression correlations with other genes in the module (i.e., connectivity).

### 4.7. Quantitative RT-PCR Validation and Expression Analysis

Total RNA isolated from the samples described above was used as a template and reverse-transcribed using the M5 Super plus qPCR RT kit (with gDNA remover) (Mei5 Biotechnology, Co., Ltd., Beijing, China). qRT-PCR was conducted using 2× M5 HiPer SYBR Premix EsTaq (with Tli RNaseH) (Mei5 Biotechnology, Co., Ltd., Beijing, China) according to the manufacturer’s recommendations with an ABI 7500 Fast RT PCR system (AB Ltd., Lincoln, NE, USA). A total of 14 DEGs were used for qRT-PCR analysis, and three technical replicates were performed for each reaction. The sequences of the primers were designed using an online tool (http://www.primer3plus.com/, accessed on 2 November 2022), and *18S rRNA* was utilized as the housekeeping gene. All the primer sequences used in this study are listed in Appendix A. The relative expression levels of each gene in the samples were calculated using the 2^−∆∆CT^ method.

### 4.8. Statistical Analysis

Statistical analyses were performed using IBM SPSS Statistics 20.0 software (IBM Inc., New York, NY, USA). Percentage data were converted to proportions (divided by 100) and arcsin square root-transformed before analysis of variance to meet heteroscedasticity assumptions. LSD multiple comparison tests were conducted to evaluate the significance of differences among treatments; the threshold for statistical significance in these tests was *p* < 0.05. Two-sample *t*-tests were performed to determine whether diploid and tetraploid plants significantly differed in basal diameter, plant height, cell wall thickness, lignin content, and hormone content.

## 5. Conclusions

After a whole-genome duplication event, the lignin content of tetraploid *P. hopeiensis* is lower than that of its isogenic diploid progenitor. Lignin biosynthesis pathways were affected by whole-genome doubling due to changes in the content of GA. Reductions in the concentration of GA abolished DELLA repression, which promoted the activities of downstream proteins repressed by DELLAs. SCL14, a major repressor encoding the DELLA protein GAI in GA signaling, interacts directly with NAC043, suppressing the NAC043–MYB61 signaling cascade and inhibiting the expression of the lignin biosynthesis gene. These findings provide new insights into the regulation of lignin formation by GA. Generally, interactions between SCL14 and NAC043 mediated by GA signaling regulate lignin biosynthesis in *P. hopeiensis*.

## Figures and Tables

**Figure 1 ijms-24-05809-f001:**
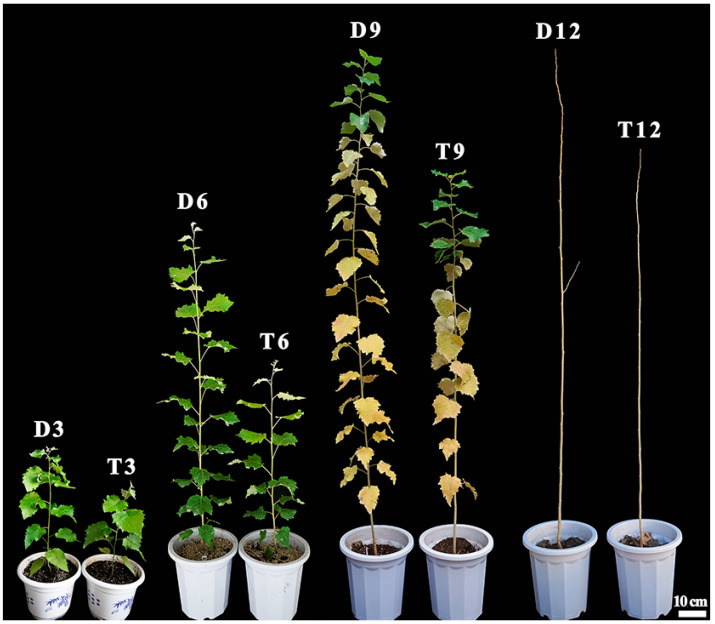
Morphological changes in diploid and tetraploid plants at different growth stages of *P. hopeiensis* clone BT1. D3, D6, D9, and D12 correspond to 3-, 6-, 9-, and 12-month-old diploid *P. hopeiensis*, respectively. T3, T6, T9, and T12 correspond to 3-, 6-, 9-, and 12-month-old tetraploid *P. hopeiensis*, respectively.

**Figure 2 ijms-24-05809-f002:**
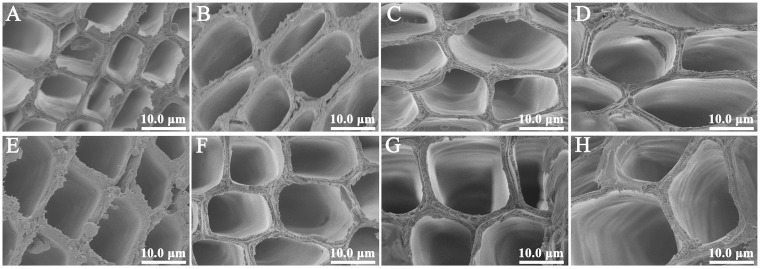
The cell wall morphology of diploid and tetraploid plants at different developmental stages of *P. hopeiensis* clone BT1. (**A**) 3-month-old diploid *P. hopeiensis*. (**B**) 6-month-old diploid *P. hopeiensis*. (**C**) 9-month-old diploid *P. hopeiensis*. (**D**) 12-month-old diploid *P. hopeiensis*. (**E**) 3-month-old tetraploid *P. hopeiensis*. (**F**) 6-month-old tetraploid *P. hopeiensis*. (**G**) 9-month-old tetraploid *P. hopeiensis*. (**H**) 12-month-old tetraploid *P. hopeiensis*.

**Figure 3 ijms-24-05809-f003:**
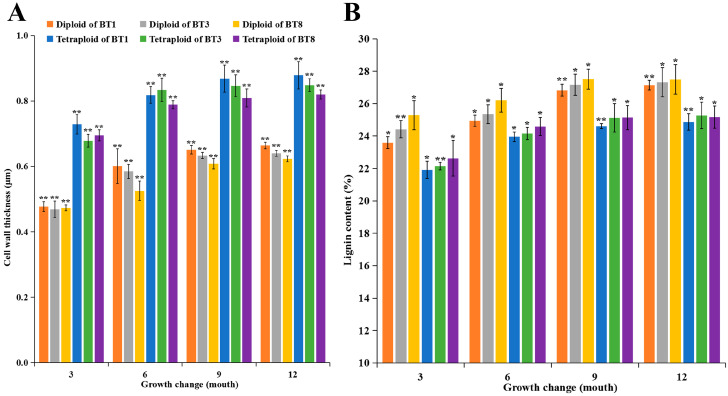
The cell wall thickness (**A**) and lignin content (**B**) of 3-, 6-, 9-, and 12-month-old diploid and tetraploid plants of *P. hopeiensis*. The vertical bars show the standard error; the asterisk indicates significant differences between diploid and tetraploid plants (* *p* < 0.05, ** *p* < 0.01).

**Figure 4 ijms-24-05809-f004:**
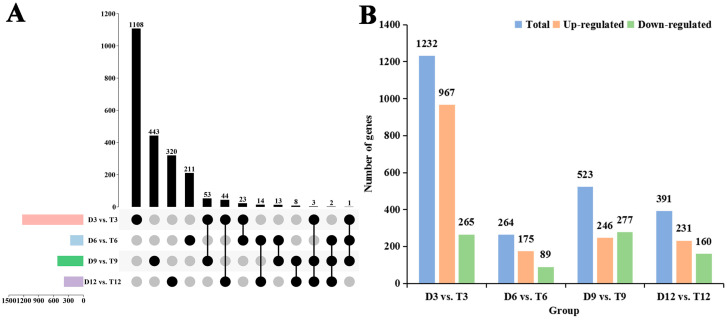
Statistics of DEGs in the diploids and tetraploids. (**A**) Upset diagram of DEGs in different comparison groups. (**B**) Statistical analysis of up-regulated and down-regulated DEGs in different comparison groups.

**Figure 5 ijms-24-05809-f005:**
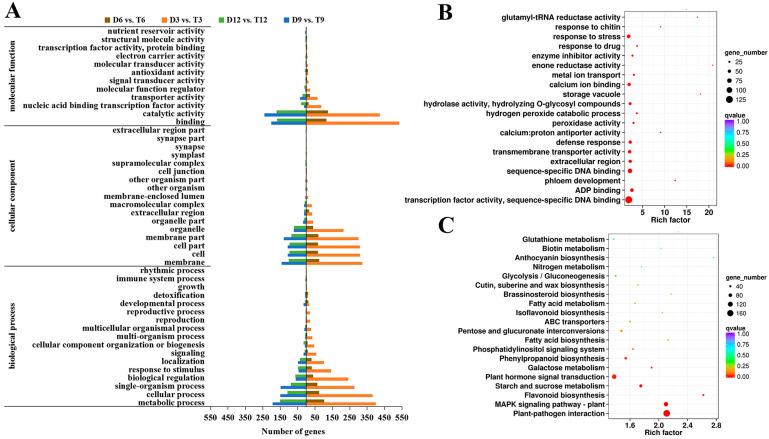
Preliminary analysis of the biological functions of DEGs from diploids and tetraploids. (**A**) Level-2 GO functional classifications of DEGs from diploids and tetraploids. (**B**) GO enrichment analysis of all DEGs from diploids and tetraploids. (**C**) KEGG enrichment analysis of all DEGs from diploids and tetraploids.

**Figure 6 ijms-24-05809-f006:**
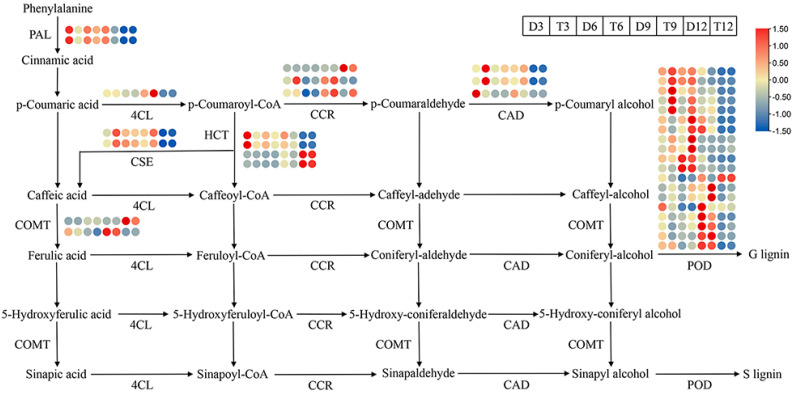
Heatmap showing the expression profiles of DEGs involved in the lignin biosynthesis pathway in diploid and tetraploid *P. hopeiensis*. The color scale shows fragments per kilobase of transcript per million mapped reads (FPKM) values, with blue colors indicating low values and red values indicating high values.

**Figure 7 ijms-24-05809-f007:**
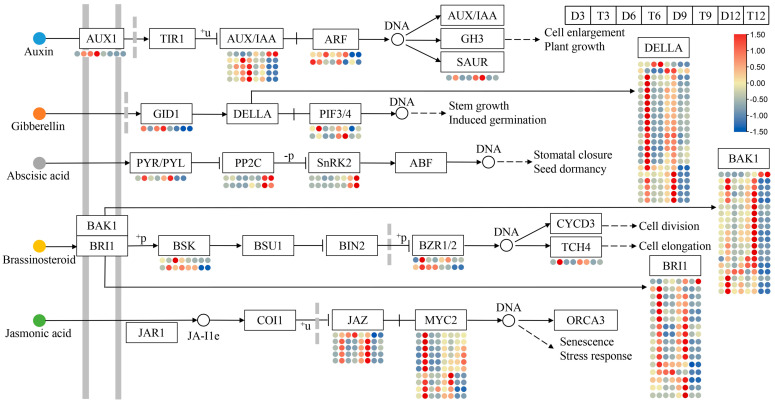
Heatmap showing the expression profiles of DEGs involved in plant hormone signal transduction in diploid and tetraploid *P. hopeiensis*. The color scale shows FPKM values, with blue colors indicating low values and red colors indicating high values.

**Figure 8 ijms-24-05809-f008:**
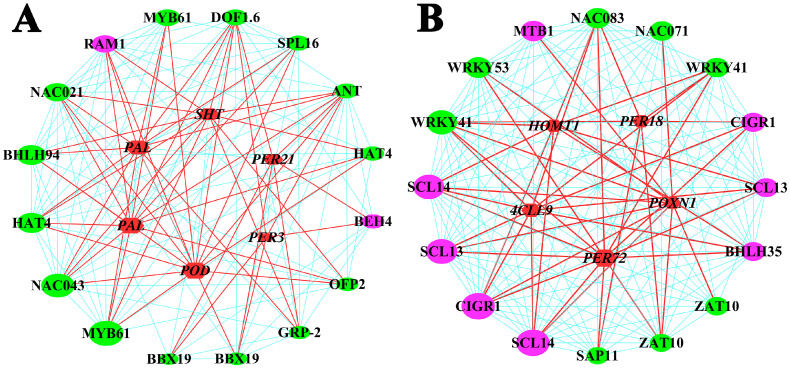
The gene co-expression network of lignin biosynthesis genes and TF genes in the brown modules (**A**) and orange modules (**B**). Red hexagonal nodes indicate lignin structural genes. Green and pink nodes indicate TFs, and pink nodes indicate TFs involved in plant hormone signal transduction. The size of the circle indicates gene connectivity, and the lines between circles indicate correlations between genes.

**Figure 9 ijms-24-05809-f009:**
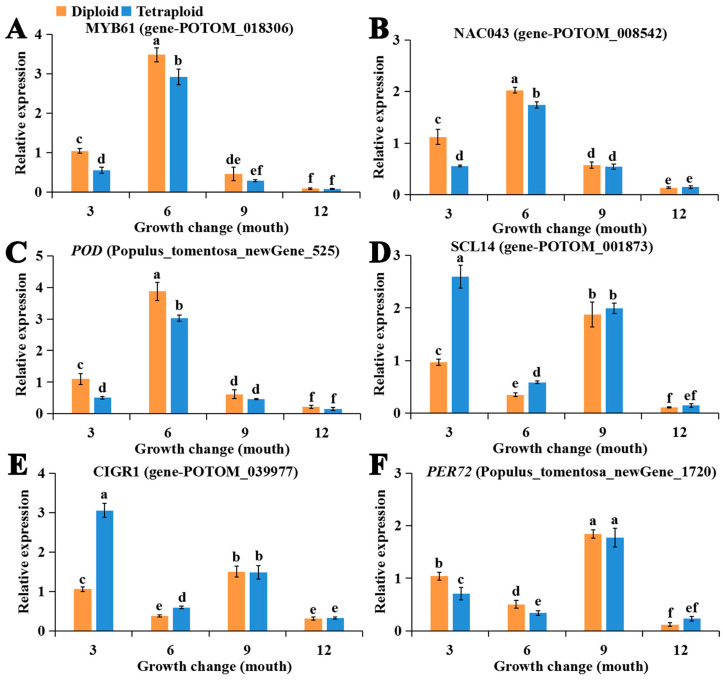
Expression levels of the six genes MYB61 (**A**), NAC043 (**B**), *POD* (**C**), SCL14 (**D**), CIGR1 (**E**), and *PER72* (**F**) according to qRT-PCR analysis. The vertical bars show the standard error; different letters above the bars indicate significant differences, and the threshold for statistical significance was *p* < 0.05.

**Figure 10 ijms-24-05809-f010:**
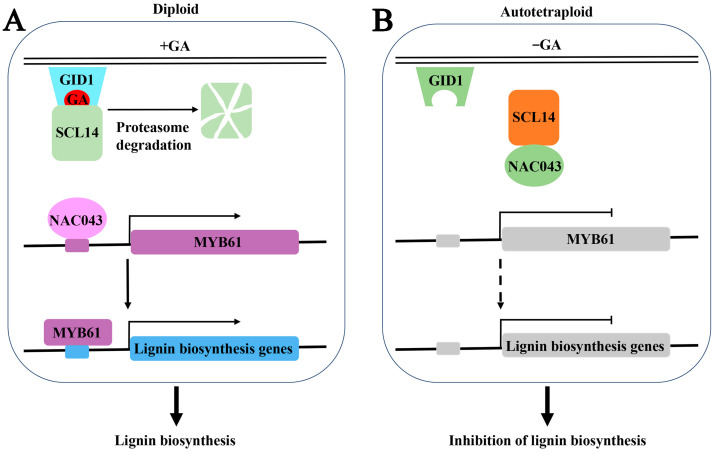
Putative model of the regulatory mechanism of lignin biosynthesis in diploid and autotetraploid *P. hopeiensis.* (**A**) In diploids, high concentrations of GA trigger the proteasomal degradation of SCL14 and frees the NAC043 to activate the expression of downstream MYB61 and lignin monomer synthase gene, which consequently promotes lignin biosynthesis. (**B**) In autotetraploids, low concentrations of GA promote the activity of SCL14. SCL14 interacts directly with NAC043, suppressing the NAC043–MYB61 signaling cascade and inhibiting lignin biosynthesis.

## Data Availability

The data presented in this study are available in Appendix A.

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
