# Peer review of "SCL14 Inhibits the Functions of the NAC043–MYB61 Signaling Cascade to Reduce the Lignin Content in Autotetraploid Populus hopeiensis"

_ijms, 2023, doi:10.3390/ijms24065809_

Round 1
Reviewer 1 Report
Dear authors,
This manuscript shows us an interesting set of results concerning the differences in lignin content between tetraploids P. hopeiensis and their diploid counterparts showing that lignin biosynthesis pathways are affected by whole-genome doubling due to changes in the content of GA, which as the authors mentioned might have implications for manipulating lignin production. Writing is very efficient, and the results are presented in a clear way. However, I would have liked to have seen, in terms of percentage increase/decrease, the quantification of some of the existing differences (e.g. lignin content, hormones content). I also advise the authors to implement some of the comments below mentioned before the publication.
Introduction
Line 69-82: The introduction should not have a summary of your results and conclusions. It should highlight the knowledge gaps and considering that, the hypothesis that the authors aim to explore. A synthesis of the main points that you will address in your study are acceptable in this section, but not the summary of your results. Please rewrite accordingly.
Results
Line 86-88: This is M&M information, so I suggest you remove it from here.
Line 119-122: The figure caption is quite confusing. Please rewrite it to make it clearer.
Line 127-144: In my opinion, it would be more interesting if you could present some percentages of increase/decrease for some of the significant changes mentioned.
Line 146.152: I cannot see the relevance of displaying so many decimal places. In my opinion, one would be enough. I leave it for your consideration.
Line 214: “this might be related to…” : I think that idea would be a better part of the discussion
Line 254-255: I think that idea would be a better part of the discussion.
Figures 3 and 9: in the xx axes please change growth age (mouth) to growth change (month).
Discussion
Line 325-351: please remove bold
General: Please mention the relevant figures whenever you are describing your results. It makes easier to the reader to observe what you are discussion
M&M
Section 4.7: Please indicate the housekeeping genes used
Reviewer 2 Report
Comments to authors:
1. Fig.8 alignment needed to be adjusted to give a full vision of the co-expression network.
2. In results section of the manuscript can be cited with appropriate citations to support the statement, for example for statements 265-267 “These two hub genes were highly co-expressed with POD (Populus_tomentosa_new Gene_525), which is involved in lignin biosynthesis.”
3. Lines 325 to 351 in the discussion needed to be changed to a uniform font size.
4. Similar to the results, some of the statements in the discussion parts needed to be supported with suitable references for example the statement “In this study, the expression of one GID1 gene (CXE15) was down-regulated in tetraploids, and 20 DELLA genes were highly expressed in tetraploids, which suggests that reductions in the concentration of GA abolished DELLA repression, promoted the activities of downstream proteins repressed by DELLAs, and inhibited lignin biosynthesis.”
